# Improved Dropout for Shallow and Deep Learning

**Zhe Li[1], Boqing Gong[2], Tianbao Yang[1]**
[1]The University of Iowa, Iowa city, IA 52245
[2]University of Central Florida, Orlando, FL 32816
{zhe-li-1,tianbao-yang}@uiowa.edu
bgong@crcv.ucf.edu

## Abstract

Dropout has been witnessed with great success in training deep neural networks by independently zeroing out the outputs of neurons at random. It has also received a surge of interest for shallow learning, e.g., logistic regression. However, the independent sampling for dropout could be suboptimal for the sake of convergence. In this paper, we propose to use multinomial sampling for dropout, i.e., sampling features or neurons according to a multinomial distribution with different probabilities for different features/neurons. To exhibit the optimal dropout probabilities, we analyze the shallow learning with multinomial dropout and establish the risk bound for stochastic optimization. By minimizing a sampling dependent factor in the risk bound, we obtain a distribution-dependent dropout with sampling probabilities dependent on the second order statistics of the data distribution. To tackle the issue of evolving distribution of neurons in deep learning, we propose an efficient adaptive dropout (named **evolutional dropout**) that computes the sampling probabilities on-the-fly from a mini-batch of examples. Empirical studies on several benchmark datasets demonstrate that the proposed dropouts achieve not only much faster convergence and but also a smaller testing error than the standard dropout. For example, on the CIFAR-100 data, the evolutional dropout achieves relative improvements over 10% on the prediction performance and over 50% on the convergence speed compared to the standard dropout.

## 1 Introduction

Dropout has been widely used to avoid overfitting of deep neural networks with a large number of parameters [9, 16], which usually identically and independently at random samples neurons and sets their outputs to be zeros. Extensive experiments [4] have shown that dropout can help obtain the state-of-the-art performance on a range of benchmark data sets. Recently, dropout has also been found to improve the performance of logistic regression and other single-layer models for natural language tasks such as document classification and named entity recognition [21].

In this paper, instead of identically and independently at random zeroing out features or neurons, we propose to use multinomial sampling for dropout, i.e., sampling features or neurons according to a multinomial distribution with different probabilities for different features/neurons. Intuitively, it makes more sense to use non-uniform multinomial sampling than identical and independent sampling for different features/neurons. For example, in shallow learning if input features are centered, we can drop out features with small variance more frequently or completely allowing the training to focus on more important features and consequentially enabling faster convergence. To justify the multinomial sampling for dropout and reveal the optimal sampling probabilities, we conduct a rigorous analysis on the risk bound of shallow learning by stochastic optimization with multinomial dropout, and demonstrate that a distribution-dependent dropout leads to a smaller expected risk (i.e., faster convergence and smaller generalization error).

Inspired by the distribution-dependent dropout, we propose a data-dependent dropout for shallow learning, and an evolutional dropout for deep learning. For shallow learning, the sampling probabilities are computed from the second order statistics of features of the training data. For deep learning, the sampling probabilities of dropout for a layer are computed on-the-fly from the second-order statistics of the layer's outputs based on a mini-batch of examples. This is particularly suited for deep learning because (i) the distribution of each layer's outputs is evolving over time, which is known as internal covariate shift [5]; (ii) passing through all the training data in deep neural networks (in particular deep convolutional neural networks) is much more expensive than through a mini-batch of examples. For a mini-batch of examples, we can leverage parallel computing architectures to accelerate the computation of sampling probabilities.

We note that the proposed evolutional dropout achieves similar effect to the batch normalization technique (Z-normalization based on a mini-batch of examples) [5] but with different flavors. Both approaches can be considered to tackle the issue of internal covariate shift for accelerating the convergence. Batch normalization tackles the issue by normalizing the output of neurons to zero mean and unit variance and then performing dropout independently [1]. In contrast, our proposed evolutional dropout tackles this issue from another perspective by exploiting a distribution-dependent dropout, which adapts the sampling probabilities to the evolving distribution of a layer's outputs. In other words, it uses normalized sampling probabilities based on the second order statistics of internal distributions. Indeed, we notice that for shallow learning with Z-normalization (normalizing each feature to zero mean and unit variance) the proposed data-dependent dropout reduces to uniform dropout that acts similarly to the standard dropout. Because of this connection, the presented theoretical analysis also sheds some lights on the power of batch normalization from the angle of theory. Compared to batch normalization, the proposed distribution-dependent dropout is still attractive because (i) it is rooted in theoretical analysis of the risk bound; (ii) it introduces no additional parameters and layers without complicating the back-propagation and the inference; (iii) it facilitates further research because its shares the same mathematical foundation as standard dropout (e.g., equivalent to a form of data-dependent regularizer) [18].

We summarize the main contributions of the paper below.

- We propose a multinomial dropout and demonstrate that a distribution-dependent dropout leads to a faster convergence and a smaller generalization error through the risk bound analysis for shallow learning.
- We propose an efficient evolutional dropout for deep learning based on the distribution-dependent dropout.
- We justify the proposed dropouts for both shallow learning and deep learning by experimental results on several benchmark datasets.

In the remainder, we first review some related work and preliminaries. We present the main results in Section 4 and experimental results in Section 5.

## 2  Related Work

In this section, we review some related work on dropout and optimization algorithms for deep learning.

Dropout is a simple yet effective technique to prevent overfitting in training deep neural networks [16]. It has received much attention recently from researchers to study its practical and theoretical properties. Notably, Wager et al. [18], Baldi and Sadowski [2] have analyzed the dropout from a theoretical viewpoint and found that dropout is equivalent to a data-dependent regularizer. The most simple form of dropout is to multiply hidden units by i.i.d Bernoulli noise. Several recent works also found that using other types of noise works as well as Bernoulli noise (e.g., Gaussian noise), which could lead to a better approximation of the marginalized loss [20, 7]. Some works tried to optimize the hyper-parameters that define the noise level in a Bayesian framework [23, 7]. Graham et al. [3] used the same noise across a batch of examples in order to speed up the computation. The adaptive dropout proposed in[1] overlays a binary belief network over a neural netowrk, incurring more computational overhead to dropout because one has to train the additional binary belief network. In constrast,

the present work proposes a new dropout with noise sampled according to distribution-dependent sampling probabilities. To the best of our knowledge, this is the first work that rigorously studies this type of dropout with theoretical analysis of the risk bound. It is demonstrated that the new dropout can improve the speed of convergence.

Stochastic gradient descent with back-propagation has been used a lot in optimizing deep neural networks. However, it is notorious for its slow convergence especially for deep learning. Recently, there emerge a battery of studies trying to accelearte the optimization of deep learning [17, 12, 22, 5, 6], which tackle the problem from different perspectives. Among them, we notice that the developed evolutional dropout for deep learning achieves similar effect as batch normalization [5] addressing the internal covariate shift issue (i.e., evolving distributions of internal hidden units).

## 3 Preliminaries

In this section, we present some preliminaries, including the framework of risk minimization in machine learning and learning with dropout noise. We also introduce the multinomial dropout, which allows us to construct a distribution-dependent dropout as revealed in the next section.

Let $(\mathbf{x}, y)$ denote a feature vector and a label, where $\mathbf{x} \in \mathbb{R}^d$ and $y \in \mathcal{Y}$. Denote by $\mathcal{P}$ the joint distribution of $(\mathbf{x}, y)$ and denote by $\mathcal{D}$ the marginal distribution of $\mathbf{x}$. The goal of risk minimization is to learn a prediction function $f(\mathbf{x})$ that minimizes the expected loss, i.e., $\min_{f \in \mathcal{H}} \mathrm{E}_{\mathcal{P}}[\ell(f(\mathbf{x}), y)]$, where $\ell(z, y)$ is a loss function (e.g., the logistic loss) that measures the inconsistency between $z$ and $y$ and $\mathcal{H}$ is a class of prediction functions. In deep learning, the prediction function $f(\mathbf{x})$ is determined by a deep neural network. In shallow learning, one might be interested in learning a linear model $f(\mathbf{x}) = \mathbf{w}^\top \mathbf{x}$. In the following presentation, the analysis will focus on the risk minimization of a linear model, i.e.,

$$\min_{\mathbf{w} \in \mathbb{R}^d} \mathcal{L}(\mathbf{w}) \triangleq \mathrm{E}_{\mathcal{P}}[\ell(\mathbf{w}^\top \mathbf{x}, y)] \tag{1}$$

In this paper, we are interested in learning with dropout, i.e., the feature vector $\mathbf{x}$ is corrupted by a dropout noise. In particular, let $\boldsymbol{\epsilon} \sim \mathcal{M}$ denote a dropout noise vector of dimension $d$, and the corrupted feature vector is given by $\widehat{\mathbf{x}} = \mathbf{x} \circ \boldsymbol{\epsilon}$, where the operator $\circ$ represents the element-wise multiplication. Let $\widehat{\mathcal{P}}$ denote the joint distribution of the new data $(\widehat{\mathbf{x}}, y)$ and $\widehat{\mathcal{D}}$ denote the marginal distribution of $\widehat{\mathbf{x}}$. With the corrupted data, the risk minimization becomes

$$\min_{\mathbf{w} \in \mathbb{R}^d} \widehat{\mathcal{L}}(\mathbf{w}) \triangleq \mathrm{E}_{\widehat{\mathcal{P}}}[\ell(\mathbf{w}^\top (\mathbf{x} \circ \boldsymbol{\epsilon}), y)] \tag{2}$$

In standard dropout [18, 4], the entries of the noise vector $\boldsymbol{\epsilon}$ are sampled independently according to $\Pr(\epsilon_j = 0) = \delta$ and $\Pr(\epsilon_j = \frac{1}{1-\delta}) = 1 - \delta$, i.e., features are dropped with a probability $\delta$ and scaled by $\frac{1}{1-\delta}$ with a probability $1 - \delta$. We can also write $\epsilon_j = \frac{b_j}{1-\delta}$, where $b_j \in \{0, 1\}, j \in [d]$ are i.i.d Bernoulli random variables with $\Pr(b_j = 1) = 1 - \delta$. The scaling factor $\frac{1}{1-\delta}$ is added to ensure that $\mathrm{E}_{\boldsymbol{\epsilon}}[\widehat{\mathbf{x}}] = \mathbf{x}$. It is obvious that using the standard dropout different features will have equal probabilities to be dropped out or to be selected independently. However, in practice some features could be more informative than the others for learning purpose. Therefore, it makes more sense to assign different sampling probabilities for different features and make the features compete with each other.

To this end, we introduce the following multinomial dropout.

**Definition 1.** *(Multinomial Dropout) A multinomial dropout is defined as $\widehat{\mathbf{x}} = \mathbf{x} \circ \boldsymbol{\epsilon}$, where $\epsilon_i = \frac{m_i}{k p_i}, i \in [d]$ and $\{m_1, \ldots, m_d\}$ follow a multinomial distribution $Mult(p_1, \ldots, p_d; k)$ with $\sum_{i=1}^d p_i = 1$ and $p_i \geq 0$.*

**Remark:** The multinomial dropout allows us to use non-uniform sampling probabilities $p_1, \ldots, p_d$ for different features. The value of $m_i$ is the number of times that the $i$-th feature is selected in $k$ independent trials of selection. In each trial, the probability that the $i$-th feature is selected is given by $p_i$. As in the standard dropout, the normalization by $k p_i$ is to ensure that $\mathrm{E}_{\boldsymbol{\epsilon}}[\widehat{\mathbf{x}}] = \mathbf{x}$. The parameter $k$ plays the same role as the parameter $1 - \delta$ in standard dropout, which controls the number of features to be dropped. In particular, the expected total number of the kept features using multinomial dropout is $k$ and that using standard dropout is $d(1 - \delta)$. In the sequel, to make fair comparison between

the two dropouts, we let $k = d(1 - \delta)$. In this case, when a uniform distribution $p_i = 1/d$ is used in multinomial dropout to which we refer as *uniform dropout*, then $\epsilon_i = \frac{m_i}{1-\delta}$, which acts similarly to the standard dropout using i.i.d Bernoulli random variables. Note that another choice to make the sampling probabilities different is still using i.i.d Bernoulli random variables but with different probabilities for different features. However, multinomial dropout is more suitable because (i) it is easy to control the level of dropout by varying the value of $k$; (ii) it gives rise to natural competition among features because of the constraint $\sum_i p_i = 1$; (iii) it allows us to minimize the sampling dependent risk bound for obtaining a better distribution than uniform sampling.

**Dropout is a data-dependent regularizer**   Dropout as a regularizer has been studied in [18, 2] for logistic regression, which is stated in the following proposition for ease of discussion later.

**Proposition 1.** *If* $\ell(z, y) = \log(1 + \exp(-yz))$, *then*

$$\mathrm{E}_{\widehat{\mathcal{P}}}[\ell(\mathbf{w}^\top \widehat{\mathbf{x}}, y)] = \mathrm{E}_{\mathcal{P}}[\ell(\mathbf{w}^\top \mathbf{x}, y)] + R_{\mathcal{D}, \mathcal{M}}(\mathbf{w}) \tag{3}$$

*where* $\mathcal{M}$ *denotes the distribution of* $\epsilon$ *and* $R_{\mathcal{D}, \mathcal{M}}(\mathbf{w}) = \mathrm{E}_{\mathcal{D}, \mathcal{M}} \left[ \log \frac{\exp(\mathbf{w}^\top \frac{\mathbf{x} \circ \epsilon}{2}) + \exp(-\mathbf{w}^\top \frac{\mathbf{x} \circ \epsilon}{2})}{\exp(\mathbf{w}^\top \mathbf{x}/2) + \exp(-\mathbf{w}^\top \mathbf{x}/2)} \right]$.

**Remark:** It is notable that $R_{\mathcal{D}, \mathcal{M}} \geq 0$ due to the Jensen inequality. Using the second order Taylor expansion, [18] showed that the following approximation of $R_{\mathcal{D}, \mathcal{M}}(\mathbf{w})$ is easy to manipulate and understand:

$$\widehat{R}_{\mathcal{D}, \mathcal{M}}(\mathbf{w}) = \frac{\mathrm{E}_{\mathcal{D}}[q(\mathbf{w}^\top \mathbf{x})(1 - q(\mathbf{w}^\top \mathbf{x}))\mathbf{w}^\top C_{\mathcal{M}}(\mathbf{x} \circ \epsilon)\mathbf{w}]}{2} \tag{4}$$

where $q(\mathbf{w}^\top \mathbf{x}) = \frac{1}{1 + \exp(-\mathbf{w}^\top \mathbf{x}/2)}$, and $C_{\mathcal{M}}$ denotes the covariance matrix in terms of $\epsilon$. In particular, if $\epsilon$ is the standard dropout noise, then $C_{\mathcal{M}}[\mathbf{x} \circ \epsilon] = diag(x_1^2 \delta/(1 - \delta), \ldots, x_d^2 \delta/(1 - \delta))$, where $diag(s_1, \ldots, s_n)$ denotes a $d \times d$ diagonal matrix with the $i$-th entry equal to $s_i$. If $\epsilon$ is the multinomial dropout noise in Definition 1, we have

$$C_{\mathcal{M}}[\mathbf{x} \circ \epsilon] = \frac{1}{k} diag(x_i^2/p_i) - \frac{1}{k}\mathbf{x}\mathbf{x}^\top \tag{5}$$

# 4   Learning with Multinomial Dropout

In this section, we analyze a stochastic optimization approach for minimizing the dropout loss in (2). Assume the sampling probabilities are known. We first obtain a risk bound of learning with multinomial dropout for stochastic optimization. Then we try to minimize the factors in the risk bound that depend on the sampling probabilities. We would like to emphasize that our goal here is not to show that using dropout would render a smaller risk than without using dropout, but rather focus on the impact of different sampling probabilities on the risk. Let the initial solution be $\mathbf{w}_1$. At the iteration $t$, we sample $(\mathbf{x}_t, y_t) \sim \mathcal{P}$ and $\epsilon_t \sim \mathcal{M}$ as in Definition 1 and then update the model by

$$\mathbf{w}_{t+1} = \mathbf{w}_t - \eta_t \nabla \ell(\mathbf{w}_t^\top (\mathbf{x}_t \circ \epsilon_t), y_t) \tag{6}$$

where $\nabla \ell$ denotes the (sub)gradient in terms of $\mathbf{w}_t$ and $\eta_t$ is a step size. Suppose we run the stochastic optimization by $n$ steps (i.e., using $n$ examples) and compute the final solution as $\widehat{\mathbf{w}}_n = \frac{1}{n}\sum_{t=1}^n \mathbf{w}_t$.

We note that another approach of learning with dropout is to minimize the empirical risk by marginalizing out the dropout noise, i.e., replacing the true expectations $\mathrm{E}_{\mathcal{P}}$ and $\mathrm{E}_{\mathcal{D}}$ in (3) with empirical expectations over a set of samples $(\mathbf{x}_1, y_1), \ldots, (\mathbf{x}_n, y_n)$ denoted by $\mathrm{E}_{\mathcal{P}_n}$ and $\mathrm{E}_{\mathcal{D}_n}$. Since the data dependent regularizer $R_{\mathcal{D}_n, \mathcal{M}}(\mathbf{w})$ is difficult to compute, one usually uses an approximation $\widehat{R}_{\mathcal{D}_n, \mathcal{M}}(\mathbf{w})$ (e.g., as in (4)) in place of $R_{\mathcal{D}_n, \mathcal{M}}(\mathbf{w})$. However, the resulting problem is a non-convex optimization, which together with the approximation error would make the risk analysis much more involved. In contrast, the update in (6) can be considered as a stochastic gradient descent update for solving the convex optimization problem in (2), allowing us to establish the risk bound based on previous results of stochastic gradient descent for risk minimization [14, 15]. Nonetheless, this restriction does not lose the generality. Indeed, stochastic optimization is usually employed for solving empirical loss minimization in big data and deep learning.

The following theorem establishes a risk bound of $\widehat{\mathbf{w}}_n$ in expectation.

**Theorem 1.** *Let $\mathcal{L}(\mathbf{w})$ be the expected risk of $\mathbf{w}$ defined in (1). Assume $\mathrm{E}_{\widehat{\mathcal{D}}}[\|\mathbf{x} \circ \boldsymbol{\epsilon}\|_2^2] \leq B^2$ and $\ell(z, y)$ is G-Lipschitz continuous. For any $\|\mathbf{w}_*\|_2 \leq r$, by appropriately choosing $\eta$, we can have*

$$\mathrm{E}[\mathcal{L}(\widehat{\mathbf{w}}_n) + R_{\mathcal{D},\mathcal{M}}(\widehat{\mathbf{w}}_n)] \leq \mathcal{L}(\mathbf{w}_*) + R_{\mathcal{D},\mathcal{M}}(\mathbf{w}_*) + \frac{GBr}{\sqrt{n}}$$

*where $\mathrm{E}[\cdot]$ is taking expectation over the randomness in $(\mathbf{x}_t, y_t, \boldsymbol{\epsilon}_t), t = 1, \ldots, n$.*

**Remark:** In the above theorem, we can choose $\mathbf{w}_*$ to be the best model that minimizes the expected risk in (1). Since $R_{\mathcal{D},M}(\mathbf{w}) \geq 0$, the upper bound in the theorem above is also the upper bound of the risk of $\widehat{\mathbf{w}}_n$, i.e., $\mathcal{L}(\widehat{\mathbf{w}}_n)$, in expectation. The proof of the above theorem follows the standard analysis of stochastic gradient descent. The detailed proof of theorem is included in the appendix.

### 4.1 Distribution Dependent Dropout

Next, we consider the sampling dependent factors in the risk bounds. From Theorem 1, we can see that there are two terms that depend on the sampling probabilities, i.e., $B^2$ - the upper bound of $\mathrm{E}_{\widehat{\mathcal{D}}}[\|\mathbf{x} \circ \boldsymbol{\epsilon}\|_2^2]$, and $R_{\mathcal{D},\mathcal{M}}(\mathbf{w}_*) - R_{\mathcal{D},\mathcal{M}}(\widehat{\mathbf{w}}_n) \leq R_{\mathcal{D},\mathcal{M}}(\mathbf{w}_*)$. We note that the second term also depends on $\mathbf{w}_*$ and $\widehat{\mathbf{w}}_n$, which is more difficult to optimize. We first try to minimize $\mathrm{E}_{\widehat{\mathcal{D}}}[\|\mathbf{x} \circ \boldsymbol{\epsilon}\|_2^2]$ and present the discussion on minimizing $R_{\mathcal{D},\mathcal{M}}(\mathbf{w}_*)$ later. From Theorem 1, we can see that minimizing $\mathrm{E}_{\widehat{\mathcal{D}}}[\|\mathbf{x} \circ \boldsymbol{\epsilon}\|_2^2]$ would lead to not only a smaller risk (given the same number of total examples, smaller $\mathrm{E}_{\widehat{\mathcal{D}}}[\|\mathbf{x} \circ \boldsymbol{\epsilon}\|_2^2]$ gives a smaller risk bound) but also a faster convergence (with the same number of iterations, smaller $\mathrm{E}_{\widehat{\mathcal{D}}}[\|\mathbf{x} \circ \boldsymbol{\epsilon}\|_2^2]$ gives a smaller optimization error).

Due to the limited space, the proofs of Proposition 2, 3, 4 are included in supplement. The following proposition simplifies the expectation $\mathrm{E}_{\widehat{\mathcal{D}}}[\|\mathbf{x} \circ \boldsymbol{\epsilon}\|_2^2]$.

**Proposition 2.** *Let $\boldsymbol{\epsilon}$ follow the distribution $\mathcal{M}$ defined in Definition 1. Then*

$$\mathrm{E}_{\widehat{\mathcal{D}}}[\|\mathbf{x} \circ \boldsymbol{\epsilon}\|_2^2] = \frac{1}{k} \sum_{i=1}^{d} \frac{1}{p_i} \mathrm{E}_{\mathcal{D}}[x_i^2] + \frac{k-1}{k} \sum_{i=1}^{d} \mathrm{E}_{\mathcal{D}}[x_i^2] \tag{7}$$

Given the expression of $\mathrm{E}_{\widehat{\mathcal{D}}}[\|\mathbf{x} \circ \boldsymbol{\epsilon}\|_2^2]$ in Proposition 2, we can minimize it over $\mathbf{p}$, leading to the following result.

**Proposition 3.** *The solution to $\mathbf{p}_* = \arg\min_{\mathbf{p} \geq 0, \mathbf{p}^\top \mathbf{1} = 1} \mathrm{E}_{\widehat{\mathcal{D}}}[\|\mathbf{x} \circ \boldsymbol{\epsilon}\|_2^2]$ is given by*

$$p_i^* = \frac{\sqrt{\mathrm{E}_{\mathcal{D}}[x_i^2]}}{\sum_{j=1}^{d} \sqrt{\mathrm{E}_{\mathcal{D}}[x_j^2]}}, i = 1, \ldots, d \tag{8}$$

Next, we examine $R_{\mathcal{D},\mathcal{M}}(\mathbf{w}_*)$. Since direct manipulation on $R_{\mathcal{D},\mathcal{M}}(\mathbf{w}_*)$ is difficult, we try to minimize the second order Taylor expansion $\widehat{R}_{\mathcal{D},\mathcal{M}}(\mathbf{w}_*)$ for logistic loss. The following theorem establishes an upper bound of $\widehat{R}_{\mathcal{D},\mathcal{M}}(\mathbf{w}_*)$.

**Proposition 4.** *Let $\boldsymbol{\epsilon}$ follow the distribution $\mathcal{M}$ defined in Definition 1. We have $\widehat{R}_{\mathcal{D},\mathcal{M}}(\mathbf{w}_*) \leq \frac{1}{8k}\|\mathbf{w}_*\|_2^2 \left( \sum_{i=1}^{d} \frac{\mathrm{E}_{\mathcal{D}}[\mathbf{x}_i^2]}{p_i} - \mathrm{E}_{\mathcal{D}}[\|\mathbf{x}\|_2^2] \right)$*

**Remark:** By minimizing the relaxed upper bound in Proposition 4, we obtain the same sampling probabilities as in (8). We note that a tighter upper bound can be established, however, which will yield sampling probabilities dependent on the unknown $\mathbf{w}_*$.

In summary, using the probabilities in (8), we can reduce both $\mathrm{E}_{\widehat{\mathcal{D}}}[\|\mathbf{x} \circ \boldsymbol{\epsilon}\|_2^2]$ and $R_{\mathcal{D},\mathcal{M}}(\mathbf{w}_*)$ in the risk bound, leading to a faster convergence and a smaller generalization error. In practice, we can use empirical second-order statistics to compute the probabilities, i.e.,

$$p_i = \frac{\sqrt{\frac{1}{n} \sum_{j=1}^{n} [[\mathbf{x}_j]_i^2]}}{\sum_{i'=1}^{d} \sqrt{\frac{1}{n} \sum_{j=1}^{n} [[\mathbf{x}_j]_{i'}^2]}} \tag{9}$$

where $[\mathbf{x}_j]_i$ denotes the $i$-th feature of the $j$-th example, which gives us a data-dependent dropout. We state it formally in the following definition.

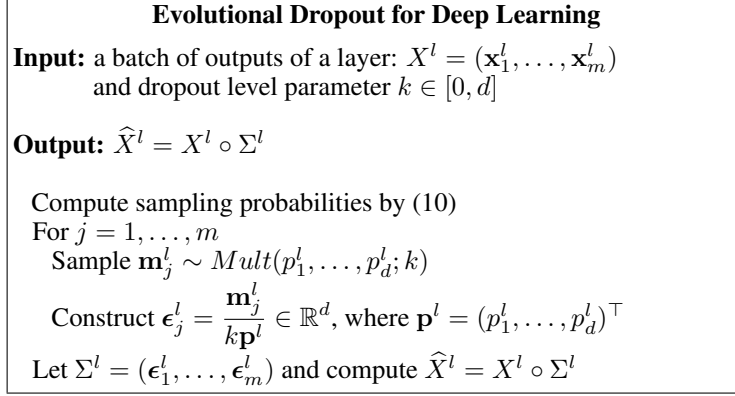

**Evolutional Dropout for Deep Learning**

**Input:** a batch of outputs of a layer: $X^l = (\mathbf{x}_1^l, \ldots, \mathbf{x}_m^l)$
and dropout level parameter $k \in [0, d]$

**Output:** $\widehat{X}^l = X^l \circ \Sigma^l$

  Compute sampling probabilities by (10)
  For $j = 1, \ldots, m$
    Sample $\mathbf{m}_j^l \sim Mult(p_1^l, \ldots, p_d^l; k)$
    Construct $\boldsymbol{\epsilon}_j^l = \dfrac{\mathbf{m}_j^l}{k\mathbf{p}^l} \in \mathbb{R}^d$, where $\mathbf{p}^l = (p_1^l, \ldots, p_d^l)^\top$
  Let $\Sigma^l = (\boldsymbol{\epsilon}_1^l, \ldots, \boldsymbol{\epsilon}_m^l)$ and compute $\widehat{X}^l = X^l \circ \Sigma^l$

Figure 1: Evolutional Dropout applied to a layer over a mini-batch

**Definition 2.** *(**Data-dependent Dropout**) Given a set of training examples $(\mathbf{x}_1, y_1), \ldots, (\mathbf{x}_n, y_n)$. A data-dependent dropout is defined as $\widehat{\mathbf{x}} = \mathbf{x} \circ \boldsymbol{\epsilon}$, where $\epsilon_i = \frac{m_i}{kp_i}, i \in [d]$ and $\{m_1, \ldots, m_d\}$ follow a multinomial distribution $Mult(p_1, \ldots, p_d; k)$ with $p_i$ given by (9).*

**Remark:** Note that if the data is normalized such that each feature has zero mean and unit variance (i.e., according to Z-normliazation), the data-dependent dropout reduces to uniform dropout. It implies that the data-dependent dropout achieves similar effect as Z-normalization plus uniform dropout. In this sense, our theoretical analysis also explains why Z-normalization usually speeds up the training [13].

## 4.2 Evolutional Dropout for Deep Learning

Next, we discuss how to implement the distribution-dependent dropout for deep learning. In training deep neural networks, the dropout is usually added to the intermediate layers (e.g., fully connected layers and convolutional layers). Let $\mathbf{x}^l = (x_1^l, \ldots, x_d^l)$ denote the outputs of the $l$-th layer (with the index of data omitted). Adding dropout to this layer is equivalent to multiplying $\mathbf{x}^l$ by a dropout noise vector $\boldsymbol{\epsilon}^l$, i.e., feeding $\widehat{\mathbf{x}}^l = \mathbf{x}^l \circ \boldsymbol{\epsilon}^l$ as the input to the next layer. Inspired by the data-dependent dropout, we can generate $\boldsymbol{\epsilon}^l$ according to a distribution given in Definition 1 with sampling probabilities $p_i^l$ computed from $\{\mathbf{x}_1^l, \ldots, \mathbf{x}_n^l\}$ similar to that (9). However, deep learning is usually trained with big data and a deep neural network is optimized by mini-batch stochastic gradient descent. Therefore, at each iteration it would be too expensive to afford the computation to pass through all examples. To address this issue, we propose to use a mini-batch of examples to calculate the second-order statistics similar to what was done in batch normalization. Let $X^l = (\mathbf{x}_1^l, \ldots, \mathbf{x}_m^l)$ denote the outputs of the $l$-th layer for a mini-batch of $m$ examples. Then we can calculate the probabilities for dropout by

$$p_i^l = \frac{\sqrt{\frac{1}{m} \sum_{j=1}^{m} [[\mathbf{x}_j^l]_i^2]}}{\sum_{i'=1}^{d} \sqrt{\frac{1}{m} \sum_{j=1}^{m} [[\mathbf{x}_j^l]_{i'}^2]}}, i = 1, \ldots, d \tag{10}$$

which define the evolutional dropout named as such because the probabilities $p_i^l$ will also evolve as the the distribution of the layer's outputs evolve. We describe the evolutional dropout as applied to a layer of a deep neural network in Figure 1.

Finally, we would like to compare the evolutional dropout with batch normalization. Similar to batch normalization, evolutional dropout can also address the internal covariate shift issue by adapting the sampling probabilities to the evolving distribution of layers' outputs. However, different from batch normalization, evolutional dropout is a randomized technique, which enjoys many benefits as standard dropout including (i) the back-propagation is simple to implement (just multiplying the gradient of $\widehat{X}^l$ by the dropout mask to get the gradient of $X^l$); (ii) the inference (i.e., testing) remains the same [2]; (iii) it is equivalent to a data-dependent regularizer with a clear mathematical explanation;

(iv) it prevents units from co-adapting of neurons, which facilitate generalization. Moreover, the evolutional dropout has its root in distribution-dependent dropout, which has theoretical guarantee to accelerate the convergence and improve the generalization for shallow learning.

## 5 Experimental Results

In the section, we present some experimental results to justify the proposed dropouts. In all experiments, we set $\delta = 0.5$ in the standard dropout and $k = 0.5d$ in the proposed dropouts for fair comparison, where $d$ represents the number of features or neurons of the layer that dropout is applied to. For the sake of clarity, we divided the experiments into three parts. In the first part, we compare the performance of the data-dependent dropout (**d-dropout)** to the standard dropout (**s-dropout**) for logistic regression. In the second part, we compare the performance of evolutional dropout (**e-dropout**) to the standard dropout for training deep convolutional neural networks. Finally, we compare e-dropout with batch normalization.

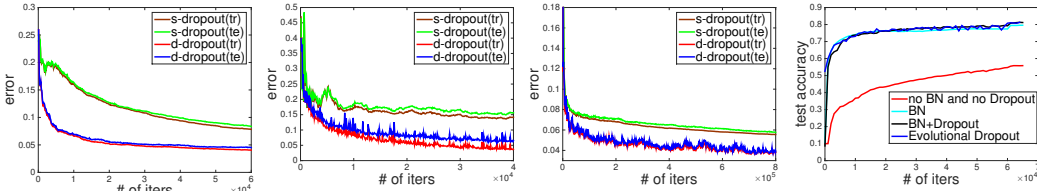

Figure 2: Left three: data-dependent dropout vs. standard dropout on three data sets (real-sim, news20, RCV1) for logistic regression; Right: Evolutional dropout vs BN on CIFAR-10. (best seen in color).

### 5.1 Shallow Learning

We implement the presented stochastic optimization algorithm. To evaluate the performance of data-dependent dropout for shallow learning, we use the three data sets: real-sim, news20 and RCV1[3]. In this experiment, we use a fixed step size and tune the step size in $[0.1, 0.05, 0.01, 0.005, 0.001, 0.0005, 0.0001]$ and report the best results in terms of convergence speed on the training data for both standard dropout and data-dependent dropout. The left three panels in Figure 2 show the obtained results on these three data sets. In each figure, we plot both the training error and the testing error. We can see that both the training and testing errors using the proposed data-dependent dropout decrease much faster than using the standard dropout and also a smaller testing error is achieved by using the data-dependent dropout.

### 5.2 Evolutional Dropout for Deep Learning

We would like to emphasize that we are not aiming to obtain better prediction performance by trying different network structures and different engineering tricks such as data augmentation, whitening, etc., but rather focus on the comparison of the proposed dropout to the standard dropout using Bernoulli noise on the same network structure. In our experiments, we use the default splitting of training and testing data in all data sets. We directly optimize the neural networks using all training images without further splitting it into a validation data to be added into the training in later stages, which explains some marginal gaps from the literature results that we observed (e.g., on CIFAR-10 compared with [19]).

We conduct experiments on four benchmark data sets for comparing e-dropout and s-dropout: MNIST [10], SVHN [11], CIFAR-10 and CIFAR-100 [8]. We use the same or similar network structure as in the literatures for the four data sets. In general, the networks consist of convolution layers, pooling layers, locally connected layers, fully connected layers, softmax layers and a cost layer. For the detailed neural network structures and their parameters, please refer to the supplementary materials. The dropout is added to some fully connected layers or locally connected layers. The rectified linear activation function is used for all neurons. All the experiments are conducted using the cuda-convnet library [4]. The training procedure is similar to [9] using mini-batch SGD with momentum (0.9). The

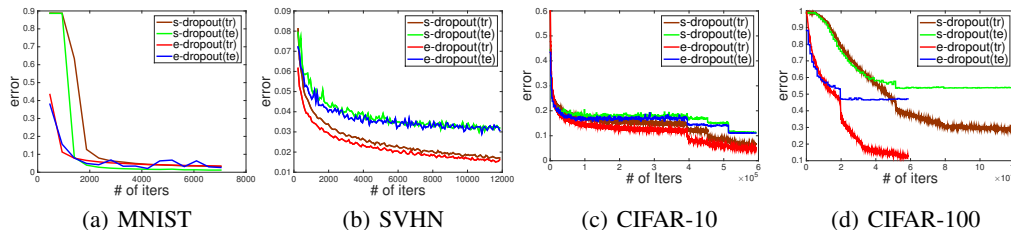

|            |          |             |              |
|------------|----------|-------------|--------------|
| (a) MNIST  | (b) SVHN | (c) CIFAR-10 | (d) CIFAR-100 |

Figure 3: Evolutional dropout vs. standard dropout on four benchmark datasets for deep learning (best seen in color).

size of mini-batch is fixed to 128. The weights are initialized based on the Gaussian distribution with mean zero and standard deviation 0.01. The learning rate (i.e., step size) is decreased after a number of epochs similar to what was done in previous works [9]. We tune the initial learning rates for s-dropout and e-dropout separately from $0.001, 0.005, 0.01, 0.1$ and report the best result on each data set that yields the fastest convergence.

Figure 3 shows the training and testing error curves in the optimization process on the four data sets using the standard dropout and the evolutional dropout. For SVHN data, we only report the first 12000 iterations, after which the error curves of the two methods almost overlap. We can see that using the evolutional dropout generally converges faster than using the standard dropout. On CIFAR-100 data, we have observed significant speed-up. In particular, the evolutional dropout achieves relative improvements over 10% on the testing performance and over 50% on the convergence speed compared to the standard dropout.

## 5.3 Comparison with the Batch Normalization (BN)

Finally, we make a comparison between the evolutional dropout and the batch normalization. For batch normalization, we use the implementation in Caffe [5]. We compare the evolutional dropout with the batch normalization on CIFAR-10 data set. The network structure is from the Caffe package and can be found in the supplement, which is different from the one used in the previous experiment. It contains three convolutional layers and one fully connected layer. Each convolutional layer is followed by a pooling layer. We compare four methods: (1) **No BN and No dropout** - without using batch normalization and dropout; (2) **BN**; (3) **BN with standard dropout**; (4) **Evolutional Dropout**. The rectified linear activation is used in all methods. We also tried BN with the sigmoid activation function, which gives worse results. For the methods with BN, three batch normalization layers are inserted before or after each pooling layer following the architecture given in Caffe package (see supplement). For the evolutional dropout training, only one layer of dropout is added to the the last convolutional layer. The mini-batch size is set to 100, the default value in Caffe. The initial learning rates for the four methods are set to the same value (0.001), and they are decreased once by ten times. The testing accuracy versus the number of iterations is plotted in the right panel of Figure 2, from which we can see that the evolutional dropout training achieves comparable performance with BN + standard dropout, which justifies our claim that evolutional dropout also addresses the internal covariate shift issue.

## 6 Conclusion

In this paper, we have proposed a distribution-dependent dropout for both shallow learning and deep learning. Theoretically, we proved that the new dropout achieves a smaller risk and faster convergence. Based on the distribution-dependent dropout, we developed an efficient evolutional dropout for training deep neural networks that adapts the sampling probabilities to the evolving distributions of layers' outputs. Experimental results on various data sets verified that the proposed dropouts can dramatically improve the convergence and also reduce the testing error.

### Acknowledgments

We thank anonymous reviewers for their comments. Z. Li and T. Yang are partially supported by National Science Foundation (IIS-1463988, IIS-1545995). B. Gong is supported in part by NSF (IIS-1566511) and a gift from Adobe.

## Footnotes

[1]The author also reported that in some cases dropout is even not necessary

[2]Different from some implementations for standard dropout which doest no scale by $1/(1-\delta)$ in training but scale by $1-\delta$ in testing, here we do scale in training and thus do not need any scaling in testing.

[3] https://www.csie.ntu.edu.tw/~cjlin/libsvmtools/datasets/

[4] https://code.google.com/archive/p/cuda-convnet/

[5]`https://github.com/BVLC/caffe/`

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
