[Supplementary Material · distribution-dropout-supplement.pdf]

# Supplement of "Improved Dropout for Shallow and Deep Learning"

**Zhe Li[1], Boqing Gong[2], Tianbao Yang[1]**
[1]The University of Iowa, Iowa city, IA 52245
[2]University of Central Florida, Orlando, FL 32816
{zhe-li-1,tianbao-yang}@uiowa.edu
bgong@crcv.ucf.edu

## 1 Proof of Theorem 1

The update given by $\mathbf{w}_{t+1} = \mathbf{w}_t - \eta \nabla \ell(\mathbf{w}_t^\top (\mathbf{x}_t \circ \boldsymbol{\epsilon}_t), y_t)$ can be considered as the stochastic gradient descent (SGD) update of the following problem

$$\min_{\mathbf{w}} \{ \widehat{\mathcal{L}}(\mathbf{w}) \triangleq \mathrm{E}_{\widehat{\mathcal{P}}}[\ell(\mathbf{w}^\top (\mathbf{x} \circ \boldsymbol{\epsilon}), y)] \}$$

Define $\mathbf{g}_t$ as $\mathbf{g}_t = \nabla \ell(\mathbf{w}_t^\top (\mathbf{x}_t \circ \boldsymbol{\epsilon}_t), y_t) = \ell'(\mathbf{w}_t^\top (\mathbf{x}_t \circ \boldsymbol{\epsilon}_t), y_t)\mathbf{x}_t \circ \boldsymbol{\epsilon}_t$, where $\ell'(z, y)$ denotes the derivative in terms of $z$. Since the loss function is $G$-Lipschitz continuous, therefore $\|\mathbf{g}_t\|_2 \leq G\|\mathbf{x}_t \circ \boldsymbol{\epsilon}_t\|_2$. According to the analysis of SGD [3], we have the following lemma.

**Lemma 1.** *Let* $\mathbf{w}_{t+1} = \mathbf{w}_t - \eta \mathbf{g}_t$ *and* $\mathbf{w}_1 = 0$. *Then for any* $\|\mathbf{w}_*\|_2 \leq r$ *we have*

$$\sum_{t=1}^{n} \mathbf{g}_t^\top (\mathbf{w}_t - \mathbf{w}_*) \leq \frac{r^2}{2\eta} + \frac{\eta}{2} \sum_{t=1}^{n} \|\mathbf{g}_t\|_2^2 \qquad (1)$$

By taking expectation on both sides over the randomness in $(\mathbf{x}_t, y_t, \boldsymbol{\epsilon}_t)$ and noting the bound on $\|\mathbf{g}_t\|_2$, we have

$$\mathrm{E}_{[n]} \left[ \sum_{t=1}^{n} \mathbf{g}_t^\top (\mathbf{w}_t - \mathbf{w}_*) \right] \leq \frac{r^2}{2\eta} + \frac{\eta}{2} \sum_{t=1}^{n} G^2 \mathrm{E}_{[n]}[\|\mathbf{x}_t \circ \boldsymbol{\epsilon}_t\|_2^2]$$

where $\mathrm{E}_{[t]}$ denote the expectation over $(\mathbf{x}_i, y_i, \boldsymbol{\epsilon}_i), i = 1, \ldots, t$. Let $\mathrm{E}_t[\cdot]$ denote the expectation over $(\mathbf{x}_t, y_t, \boldsymbol{\epsilon}_t)$ with $(\mathbf{x}_i, y_i, \boldsymbol{\epsilon}_i), i = 1, \ldots, t-1$ given. Then we have

$$\sum_{t=1}^{n} \mathrm{E}_{[t]}[\mathbf{g}_t^\top (\mathbf{w}_t - \mathbf{w}_*)] \leq \frac{r^2}{2\eta} + \frac{\eta}{2} \sum_{t=1}^{n} G^2 \mathrm{E}_t[\|\mathbf{x}_t \circ \boldsymbol{\epsilon}_t\|_2^2]$$

Since

$$\mathrm{E}_{[t]}[\mathbf{g}_t^\top (\mathbf{w}_t - \mathbf{w}_*)] = \mathrm{E}_{[t-1]}[\mathrm{E}_t[\mathbf{g}_t]^\top (\mathbf{w}_t - \mathbf{w}_*)] = \mathrm{E}_{[t-1]}[\nabla \widehat{\mathcal{L}}(\mathbf{w}_t)^\top (\mathbf{w}_t - \mathbf{w}_*)] \geq \mathrm{E}_{[t-1]}[\widehat{\mathcal{L}}(\mathbf{w}_t) - \widehat{\mathcal{L}}(\mathbf{w}_*)]$$

As a result

$$\mathrm{E}_{[n]} \left[ \sum_{t=1}^{n} (\widehat{\mathcal{L}}(\mathbf{w}_t) - \widehat{\mathcal{L}}(\mathbf{w}_*)) \right] \leq \frac{r^2}{2\eta} + \frac{\eta}{2} \sum_{t=1}^{n} G^2 \mathrm{E}_{\widehat{\mathcal{D}}}[\|\mathbf{x}_t \circ \boldsymbol{\epsilon}_t\|_2^2] \leq \frac{r^2}{2\eta} + \frac{\eta}{2} G^2 B^2 n \qquad (2)$$

where the last inequality follows the assumed upper bound of $\mathrm{E}_{\widehat{\mathcal{D}}}[\|\mathbf{x}_t \circ \boldsymbol{\epsilon}_t\|_2^2]$. Following the definition of $\widehat{\mathbf{w}}_n$ and the convexity of $\mathcal{L}(\mathbf{w})$ we have

$$\mathrm{E}_{[n]}[\widehat{\mathcal{L}}(\widehat{\mathbf{w}}_n) - \widehat{\mathcal{L}}(\mathbf{w}_*)] \leq \mathrm{E}_{[n]} \left[ \frac{1}{n} \sum_{t=1}^{n} (\widehat{\mathcal{L}}(\mathbf{w}_t) - \widehat{\mathcal{L}}(\mathbf{w}_*)) \right] \leq \frac{r^2}{2\eta n} + \frac{\eta}{2} G^2 B^2$$

By minimizing the upper bound in terms of $\eta$, we have $E_{[n]}[\widehat{\mathcal{L}}(\widehat{\mathbf{w}}_n) - \widehat{\mathcal{L}}(\mathbf{w}_*)] \leq \frac{GBr}{\sqrt{n}}$. According to Proposition 1 in the paper $\widehat{\mathcal{L}}(\mathbf{w}) = \mathcal{L}(\mathbf{w}) + R_{\mathcal{D},\mathcal{M}}(\mathbf{w})$, therefore

$$E_{[n]}[\mathcal{L}(\widehat{\mathbf{w}}_n) + R_{\mathcal{D},\mathcal{M}}(\widehat{\mathbf{w}}_n)] \leq \mathcal{L}(\mathbf{w}_*) + R_{\mathcal{D},\mathcal{M}}(\mathbf{w}_*) + \frac{GBr}{\sqrt{n}}$$

## 1.1 Proof of Lemma 1

We have the following:

$$\frac{1}{2}\|\mathbf{w}_{t+1} - \mathbf{w}_*\|_2^2 = \frac{1}{2}\|\mathbf{w}_t - \eta\mathbf{g}_t - \mathbf{w}_*\|_2^2 = \frac{1}{2}\|\mathbf{w}_t - \mathbf{w}_*\|_2^2 + \frac{\eta^2}{2}\|\mathbf{g}_t\|_2^2 - \eta(\mathbf{w}_t - \mathbf{w}_*)^\top \mathbf{g}_t$$

Then

$$(\mathbf{w}_t - \mathbf{w}_*)^\top \mathbf{g}_t \leq \frac{1}{2\eta}\|\mathbf{w}_t - \mathbf{w}_*\|_2^2 - \frac{1}{2\eta}\|\mathbf{w}_{t+1} - \mathbf{w}_*\|_2^2 + \frac{\eta}{2}\|\mathbf{g}_t\|_2^2$$

By summing the above inequality over $t = 1, \ldots, n$, we obtain

$$\sum_{t=1}^{n} \mathbf{g}_t^\top (\mathbf{w}_t - \mathbf{w}_*) \leq \frac{\|\mathbf{w}_* - \mathbf{w}_1\|_2^2}{2\eta} + \frac{\eta}{2} \sum_{t=1}^{n} \|\mathbf{g}_t\|_2^2$$

By noting that $\mathbf{w}_1 = 0$ and $\|\mathbf{w}_*\|_2 \leq r$, we obtain the inequality in Lemma 1.

## 2 Proof of Proposition 2

We have

$$E_{\widehat{\mathcal{D}}}\|\mathbf{x} \circ \boldsymbol{\epsilon}\|_2^2 = E_{\mathcal{D}} \left[ \sum_{i=1}^{d} \frac{x_i^2}{k^2 p_i^2} E[m_i^2] \right]$$

Since $\{m_1, \ldots, m_d\}$ follows a multinomial distribution $Mult(p_1, \ldots, p_d; k)$, we have

$$E[m_i^2] = var(m_i) + (E[m_i])^2 = kp_i(1 - p_i) + k^2 p_i^2$$

The result in the Proposition follows by combining the above two equations.

## 3 Proof of Proposition 3

Note that only the first term in the R.H.S of Eqn. (7) depends on $p_i$. Thus,

$$\mathbf{p}_* = \arg \min_{\mathbf{p} \geq 0, \mathbf{p}^\top \mathbf{1} = 1} \sum_{i=1}^{d} \frac{E_{\mathcal{D}}[x_i^2]}{p_i}$$

The result then follows the KKT conditions.

## 4 Proof of Proposition 4

We prove the first upper bound first. From Eqn. (4) in the paper, we have

$$\widehat{R}_{\mathcal{D},\mathcal{M}}(\mathbf{w}_*) \leq \frac{1}{8} E_{\mathcal{D}}[\mathbf{w}_*^\top C_{\mathcal{M}}(\mathbf{x} \circ \epsilon)\mathbf{w}_*]$$

where we use the fact $\sqrt{ab} \leq \frac{a+b}{2}$ for $a, b \geq 0$. Using Eqn. (5) in the paper, we have

$$E_{\mathcal{D}}[\mathbf{w}_*^\top C_{\mathcal{M}}(\mathbf{x} \circ \epsilon)\mathbf{w}_*] = E_{\mathcal{D}}\left[\mathbf{w}_*^\top \left(\frac{1}{k}diag(x_i^2/p_i) - \frac{1}{k}\mathbf{x}\mathbf{x}^\top\right)\mathbf{w}_*\right] = \frac{1}{k}E_{\mathcal{D}}\left[\sum_{i=1}^{d} \frac{w_{*i}^2 x_i^2}{p_i} - (\mathbf{w}_*^\top \mathbf{x})^2\right]$$

This gives a tight bound of $\widehat{R}_{\mathcal{D},\mathcal{M}}(\mathbf{w}_*)$, i.e.,

$$\widehat{R}_{\mathcal{D},\mathcal{M}}(\mathbf{w}_*) \leq \frac{1}{8k} \left\{ \sum_{i=1}^{d} \frac{w_{*i}^2 E_{\mathcal{D}}[\mathbf{x}_i^2]}{p_i} - E_{\mathcal{D}}(\mathbf{w}_*^\top \mathbf{x})^2 \right\}$$

By minimizing the above upper bound over $p_i$, we obtain following probabilities

$$p_i^* = \frac{\sqrt{w_{*i}^2 \mathrm{E}_{\mathcal{D}}[x_i^2]}}{\sum_{j=1}^d \sqrt{w_{*i}^2 \mathrm{E}_{\mathcal{D}}[x_j^2]}} \tag{3}$$

which depend on unknown $\mathbf{w}_*$. We address this issue, we derive a relaxed upper bound. We note that

$$C_{\mathcal{M}}(\mathbf{x} \circ \epsilon) = \mathrm{E}_{\mathcal{M}}[(\mathbf{x} \circ \epsilon - \mathbf{x})(\mathbf{x} \circ \epsilon - \mathbf{x})^\top]$$

$$\leq (\mathrm{E}_{\mathcal{M}} \|\mathbf{x} \circ \epsilon - \mathbf{x}\|_2^2) \cdot I_d = \left(\mathrm{E}_{\mathcal{M}}[\|\mathbf{x} \circ \epsilon\|_2^2] - \|\mathbf{x}\|_2^2\right) I_d$$

where $I_d$ denotes the identity matrix of dimension $d$. Thus

$$\mathrm{E}_{\mathcal{D}}[\mathbf{w}_*^\top C_{\mathcal{M}}(\mathbf{x} \circ \epsilon)\mathbf{w}_*] \leq \|\mathbf{w}_*\|_2^2 \left(\mathrm{E}_{\widehat{\mathcal{D}}}[\|\mathbf{x} \circ \epsilon\|_2^2] - \mathrm{E}_{\mathcal{D}}[\|\mathbf{x}\|_2^2]\right)$$

By noting the result in Proposition 2 in the paper, we have

$$\mathrm{E}_{\mathcal{D}}[\mathbf{w}_*^\top C_{\mathcal{M}}(\mathbf{x} \circ \epsilon)\mathbf{w}_*] \leq \frac{1}{k} \|\mathbf{w}_*\|_2^2 \left(\sum_{i=1}^d \frac{\mathrm{E}_{\mathcal{D}}[\mathbf{x}_i^2]}{p_i} - \mathrm{E}_{\mathcal{D}}[\|\mathbf{x}\|_2^2]\right)$$

which proves the upper bound in Proposition 4.

# 5   Neural Network Structures

In this section we present the neural network structures and the number of filters, filter size, padding and stride parameters for MNIST, SVHN, CIFAR-10 and CIFAR-100, respectively. Note that in Table 2, Table 3 and Table 4, the rnorm layer is the local response normalization layer and the local layer is the locally-connected layer with unshared weights.

## 5.1   MNIST

We used the similar neural network structure to [2]: two convolution layers, two fully connected layers, a softmax layer and a cost layer at the end. The dropout is added to the first fully connected layer. Tables 1 presents the neural network structures and the number of filters, filter size, padding and stride parameters for MNIST.

Table 1: The Neural Network Structure for MNIST

| Layer Type | Input Size | #Filters | Filter size | Padding/Stride | Output Size |
|---|---|---|---|---|---|
| conv1 | $28 \times 28 \times 1$ | 32 | $4 \times 4$ | 0/1 | $21 \times 21 \times 32$ |
| pool1(max) | $21 \times 21 \times 32$ | | $2 \times 2$ | 0/2 | $11 \times 11 \times 32$ |
| conv2 | $11 \times 11 \times 32$ | 64 | $5 \times 5$ | 0/1 | $7 \times 7 \times 64$ |
| pool2(max) | $7 \times 7 \times 64$ | | $3 \times 3$ | 0/3 | $3 \times 3 \times 64$ |
| fc1 | $3 \times 3 \times 64$ | | | | 150 |
| dropout | 150 | | | | 150 |
| fc2 | 150 | | | | 10 |
| softmax | 10 | | | | 10 |
| cost | 10 | | | | 1 |

## 5.2   SVHN

The neural network structure used for this data set is from [2], including 2 convolutional layers, 2 max pooling layers, 2 local response layers, 2 fully connected layers, a softmax layer and a cost layer with one dropout layer. Tables 2 presents the neural network structures and the number of filters, filter size, padding and stride parameters used for SVHN data set.

## 5.3   CIFAR-10

The neural network structure is adopted from [2], which consists two convolutional layer, two pooling layers, two local normalization response layers, 2 locally connected layers, two fully connected layers and a softmax and a cost layer. Table 3 presents the detail neural network structure and the number of filters, filter size, padding and stride parameters used.

Table 2: The Neural Network Structure for SVHN

| Layer Type | Input Size | #Filters | Filter Size | Padding/Stride | Output Size |
|---|---|---|---|---|---|
| conv1 | $28 \times 28 \times 3$ | 64 | $5 \times 5$ | 0/1 | $24 \times 24 \times 64$ |
| pool1(max) | $24 \times 24 \times 64$ | | $3 \times 3$ | 0/2 | $12 \times 12 \times 64$ |
| rnorm1 | $12 \times 12 \times 64$ | | | | $12 \times 12 \times 64$ |
| conv2 | $12 \times 12 \times 64$ | 64 | $5 \times 5$ | 2/1 | $12 \times 12 \times 64$ |
| rnorm2 | $12 \times 12 \times 64$ | | | | $12 \times 12 \times 64$ |
| pool2(max) | $12 \times 12 \times 64$ | | $3 \times 3$ | 0/2 | $6 \times 6 \times 64$ |
| local3 | $6 \times 6 \times 64$ | 64 | $3 \times 3$ | 1/1 | $6 \times 6 \times 64$ |
| local4 | $6 \times 6 \times 64$ | 32 | $3 \times 3$ | 1/1 | $6 \times 6 \times 32$ |
| dropout | 1152 | | | | 1152 |
| fc1 | 1152 | | | | 512 |
| fc10 | 512 | | | | 10 |
| softmax | 10 | | | | 10 |
| cost | 10 | | | | 1 |

Table 3: The Neural Network Structure for CIFAR-10

| Layer Type | Input Size | #Filters | Filter Size | Padding/Stride | Output Size |
|---|---|---|---|---|---|
| conv1 | $24 \times 24 \times 3$ | 64 | $5 \times 5$ | 2/1 | $24 \times 24 \times 64$ |
| pool1(max) | $24 \times 24 \times 64$ | | $3 \times 3$ | 0/2 | $12 \times 12 \times 64$ |
| rnorm1 | $12 \times 12 \times 64$ | | | | $12 \times 12 \times 64$ |
| conv2 | $12 \times 12 \times 64$ | 64 | $5 \times 5$ | 2/1 | $12 \times 12 \times 64$ |
| rnorm2 | $12 \times 12 \times 64$ | | | | $12 \times 12 \times 64$ |
| pool2(max) | $12 \times 12 \times 64$ | | $3 \times 3$ | 0/2 | $6 \times 6 \times 64$ |
| local3 | $6 \times 6 \times 64$ | 64 | $3 \times 3$ | 1/1 | $6 \times 6 \times 64$ |
| local4 | $6 \times 6 \times 64$ | 32 | $3 \times 3$ | 1/1 | $6 \times 6 \times 32$ |
| dropout | 1152 | | | | 1152 |
| fc1 | 1152 | | | | 128 |
| fc10 | 128 | | | | 10 |
| softmax | 10 | | | | 10 |
| cost | 10 | | | | 1 |

## 5.4   CIFAR-100

The network structure for this data set is similar to the neural network structure in [1], which consists of 2 convolution layers, 2 max pooling layers, 2 local response normalization layers, 2 locally connected layers, 3 fully connected layers, and a softmax and a cost layer. Table 4 presents the neural network structures and the number of filters, filter size, padding and stride parameters used for CIFAR-100 data set.

## 5.5   The Neural Network Structure used for BN

Tables 5 and 6 present the network structures of different methods in subsection 5.3 in the paper. The layer pool(ave) in Table 5 and Table 6 represents the average pooling layer.

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

Table 4: The Neural Network Structure for CIFAR-100

| Layer Type | Input Size | #Filters | Filter Size | Padding/Stride | Output Size |
|---|---|---|---|---|---|
| conv1 | $32 \times 32 \times 3$ | 64 | $5 \times 5$ | 2/1 | $32 \times 32 \times 64$ |
| pool1(max) | $32 \times 32 \times 64$ | | $3 \times 3$ | 0/2 | $16 \times 16 \times 64$ |
| rnorm1 | $16 \times 16 \times 64$ | | | | $16 \times 16 \times 64$ |
| conv2 | $16 \times 16 \times 64$ | 64 | $5 \times 5$ | 2/1 | $16 \times 16 \times 64$ |
| rnorm2 | $16 \times 16 \times 64$ | | | | $16 \times 16 \times 64$ |
| pool2(max) | $16 \times 16 \times 64$ | | $3 \times 3$ | 0/2 | $8 \times 8 \times 64$ |
| local3 | $8 \times 8 \times 64$ | 64 | $3 \times 3$ | 1/1 | $8 \times 8 \times 64$ |
| local4 | $8 \times 8 \times 64$ | 32 | $3 \times 3$ | 1/1 | $8 \times 8 \times 32$ |
| fc1 | 2048 | | | | 128 |
| dropout | 128 | | | | 128 |
| fc2 | 128 | | | | 128 |
| fc100 | 128 | | | | 100 |
| softmax | 100 | | | | 100 |
| cost | 100 | | | | 1 |

Table 5: Layers of networks for the experiment comparing with BN on CIFAR-10

| Layer Type | noBN-noDropout | BN | e-dropout |
|---|---|---|---|
| Layer 1 | conv1 | conv1 | conv1 |
| Layer 2 | pool1(max) | pool(max) | pool1(max) |
| Layer 3 | N/A | bn1 | N/A |
| Layer 4 | conv2 | conv2 | conv2 |
| Layer 5 | N/A | bn2 | N/A |
| Layer 6 | pool2(ave) | pool2(ave) | pool2(ave) |
| Layer 7 | conv3 | conv3 | conv3 |
| Layer 8 | N/A | bn3 | e-dropout |
| Layer 9 | pool3(ave) | pool3(ave) | pool3(ave) |
| Layer 10 | fc1 | fc1 | fc1 |
| Layer 11 | softmax | softmax | softmax |

Table 6: Sizes in networks for the experiment comparing with BN on CIFAR-10

| Layer Type | Input size | #Filters | Filter size | Padding/Stride | Output size |
|---|---|---|---|---|---|
| conv1 | $32 \times 32 \times 3$ | 32 | $5 \times 5$ | 2/1 | $32 \times 32 \times 32$ |
| pool1(max) | $32 \times 32 \times 32$ | | $3 \times 3$ | 0/2 | $16 \times 16 \times 32$ |
| conv2 | $16 \times 16 \times 32$ | 32 | $5 \times 5$ | 2/1 | $16 \times 16 \times 32$ |
| pool2(ave) | $16 \times 16 \times 32$ | | $3 \times 3$ | 0/2 | $8 \times 8 \times 32$ |
| conv3 | $8 \times 8 \times 32$ | 64 | $5 \times 5$ | 2/1 | $8 \times 8 \times 64$ |
| pool3(ave) | $8 \times 8 \times 64$ | | $3 \times 3$ | 0/2 | $4 \times 4 \times 64$ |
| fc1 | $4 \times 4 \times 64$ | | | | 10 |
| softmax | 10 | | | | 10 |
| cost | 10 | | | | 1 |

[3] Martin Zinkevich. Online convex programming and generalized infinitesimal gradient ascent. In *Proceedings of the International Conference on Machine Learning (ICML)*, pages 928–936, 2003.