[Reviews · NeurIPS 2016]

Reviewer 1

Summary

The paper addresses one of the recent ubiquitous techniques for improving performance of deep networks, namely dropout. The paper starts by considering a linear shallow network and develops theoretical analyses using risk bounds and update rules for using multinomial sampling for selecting the dropout neurons at each update step (using second order statistics of features of the data). Then they continue to apply the update rules for deep networks where they use second order statistics of a layer for mini-batches of the data. This also shows the connection with the internal covariate shift, well studied in the literature. The technique is showed to work much better in terms of convergence and accuracy for both shallow and deep learning on multiple datasets.

Qualitative Assessment

The paper is significant in the sense that the implementation seems to be quite simple, while the results clearly show the significant performance gain compared to standard dropout. This is shown to be the new state of the art for dropout and I expect it to be ubiquitous in a very short time. The paper seems technically sound. Claims are supported by theoretical analysis as well as practical experiments. It is a complete piece of work, starting from an initial insightful observation and then going through step by step to the final experiments showing the performance gain of their method. The paper is clearly written and well-organized, it does inform the reader and it is reproducible. The approach is quite new, in the sense that nobody has previously tackled the dropout problem in this way. Especially supported by theoretical bounds, this paves the way for more investigations and variations in this direction, i.e. considering more complex probability distributions for using dropout. Showing that this also addresses the internal covariate shift is very insightful. Minor comments: I think you should specify how costly it is. Is the convergence speed you are referring to in terms of the number of iterations (as it seems to be the case)? If so, how does it perform in terms of wall-time ? What is \mathcal{H} in line 104? Line 242 (iv) please reformulate In the proof of Theorem 1, between lines 15 and 16, second term, you forgot the widehat for \mathcal{L}. Same on line 16. In line 16 you have "the upper bound of in terms of". In proof of Lemma 1, the second equality should be an inequality. In the supplement, after line 35, second math line, first term after the inequality, you forgot the parentheses for the expectation over \mathcal{M}. In the supplementary material you say for Proposition 3 the result is following the KKT conditions, but since this is one of the main results of the paper, you could give more details here.

Confidence in this Review

2-Confident (read it all; understood it all reasonably well)


Reviewer 2

Summary

This paper proposes a new dropout scheme based on sampling features or neurons according to a multinomial distribution with different probabilities for different features or neurons. For shallow learning, it proposes a data-dependent dropout by using the second-order statistics of the features to compute the sampling probabilities. Theoretical analysis of the risk bound is provided. For deep learning, it proposes an evolutional dropout by using the second-order statistics of each layer’s output based on a mini-batch of examples (for the purpose of reducing the computational demand) to compute the sampling probabilities of dropout for that layer. Some experiments are presented to compare the proposed distribution-dependent dropout with standard dropout for both shallow and deep learning and with batch normalization.

Qualitative Assessment

TECHNICAL QUALITY It is discussed in Section 1 (lines 31-33) that features with low/zero variance can be dropped more frequently or even completely. How is this intuition supported by the theoretical analysis in Section 4 (particularly Eq. (8) or (9))? Note that the features are not automatically zero-meaned. The uniform dropout scheme described in line 135 (as a special case of multinomial dropout when all the sampling probabilities are equal) is only similar but not identical to standard dropout (the sampling probabilities for different features are not i.i.d.). It may not be good to use it in the experiments as if it is indeed the standard dropout scheme. There are some other concerns about the experiments reported. Although the emphasis of this paper is not on obtaining better prediction performance by trying different network structures and tricks (line 265-266), it aims at improving standard dropout and hence is natural to also include other improved dropout schemes as baselines in the comparative study. Such baselines should at least include the adaptive dropout method in Ba and Frey’s NIPS 2013 paper (which should have been cited) and the variational dropout method by Kingma et al. [6]. The experiments consider only one dropout rate (0.5). Different dropout rates should be reported to give a more complete picture. For shallow learning, I suggest that the author also includes a setting for s-dropout using data after Z-normalization. As noted in the remark in Section 4.1 (lines 213-217), its performance is expected to be similar to that of d-dropout. In Section 5.3, the paper concludes that e-dropout is roughly a randomized version of batch normalization plus standard dropout. This seems to imply that combining e-dropout with batch normalization will not lead to further improvement in performance. I suggest that this combination be also included in the comparison. Also, from Figure 2 (right), as the number of iterations increases the test accuracy of e-dropout fluctuates more than other methods, including BN+dropout which, like e-dropout, is also a randomized scheme. This seems to show that e-dropout is slightly worse than BN+dropout as far as learning stability is concerned. Some discussions should be provided on this. The performance gap between s-dropout and e-dropout is not always as large as that between s-dropout (without Z-normalization) and d-dropout. In terms of the final test accuracy, e-dropout does not always win. In fact, it seems to be worse than s-dropout for MNIST. NOVELTY The main theoretical result in the paper is Theorem 1. Its proof is based on standard techniques for stochastic gradient descent. The extension from shallow learning to deep learning is straightforward in that it simply uses mini-batches instead of the full data set to establish the sampling probabilities for each layer separately. No theoretical guarantee is available for the deep learning case. Nevertheless, as far as I know, this is the first data-dependent dropout method proposed with theoretical justification though only for the shallow learning case. IMPACT From the experiments, the performance gap between s-dropout and e-dropout is not always as large as that between s-dropout and d-dropout. One may question whether the improvement obtained by distribution-dependent dropout will still be significant for deeper networks. Compared with more realistic deep learning applications which need much deeper networks, I am afraid only small-scale experiments are presented for deep learning in the paper. Consequently, the potential impact that this work can bring to the deep learning community is unclear. CLARITY & PRESENTATION The paper is generally well organized and easy to read. Nevertheless, the writing has room to improve. Besides problems with English usage, there are also some language/formatting errors. Here are some examples: L22-23: “at random samples neurons and sets their outputs to be zeros” L32: “can dropout” L58: “acts similar to” L67: “a faster convergence” L73: “reminder” L80,85: seem to be errors when using citation in LaTeX L86: “to speed-up” L93-94: “developed evolutional dropout” L101: “in next section” L111: “where he operator” L135: “we refer as” L167-168: “a non-convex optimization” L172: “does not loss the generality” L179: “upper bond” L181: “The detailed proof of theorem” L185-186: “the second term also depend on” L191: “included in supplement”

Confidence in this Review

2-Confident (read it all; understood it all reasonably well)


Reviewer 3

Summary

This paper introduces a new sampling model for dropout, where different features/neurons are subjected to different multinomial distributions.

Qualitative Assessment

The paper introduces improvements to standard dropout that might be useful in practice. I particularly find Figure 2, section 5.1 and figure 3 interesting in terms of results/benchmarks.

Confidence in this Review

2-Confident (read it all; understood it all reasonably well)


Reviewer 4

Summary

This paper presents an improved dropout approach for shallow and deep learning. Instead of uniform sampling as done in the conventional dropout, the proposed approach drops the neurons according to a multi-normal distribution. It is claimed that the improved dropout performs similarly with batch normalization. A few experiments are conducted to justify the powerfulness of the proposed approach.

Qualitative Assessment

Overall, I like the ideas of data-dependent dropout and evolutionary dropout. I also think it is interesting that it performs similarly to batch normalization. What is not satisfactory is that experiments are not strong enough. It is suggested to add more results, e.g., in a table form, to compare with state-of-the-arts and results in ImageNet.

Confidence in this Review

2-Confident (read it all; understood it all reasonably well)


Reviewer 5

Summary

In this paper, the authors have proposed a distribution-dependent dropout for both shallow learning and deep learning. The authors have theoretically proved that the new dropout achieves a smaller risk and faster convergence. Based on the distribution-dependent dropout, the authors developed an efficient evolutional droupout for training deep neural networks that adapts the sampling probabilities to the evolving distributions of layers' outputs.

Qualitative Assessment

The authors propose a multinomial dropout and demonstrate that a distribution-dependent dropout leads to a faster convergence and a smaller generalization error through the risk bound analysis for shallow learning. This paper is well written and easy to follow. I have the following concerns. (1) I suspect whether the novelty of this submission meets the requirement of NIPS.The authors may want to emphsize their novelty in the rebuttal. (2) In the experiment part, why do the authors only report the best results on each dataset that yields the fastest convergence? (3) Why do the authors only compare the evolutional dropout with BN on CIFAR-10 dataset? The authors may want to do similar experiments on other datasets to verify the effectiveness of the proposed approach. Minor comments: The citation package is not correctly used. In section 2, there are some "author ?".

Confidence in this Review

2-Confident (read it all; understood it all reasonably well)


Reviewer 6

Summary

This paper present a new variant of dropout which uses separate drop probabilities for each unit. The probabilities are chosen according to the second order statistics of the units. The authors provide an in depth analysis showing that this choice of probabilities minimizes an upper bound to the expected risk, thus potentially leading to faster convergence and lower final error. They also provide some empirical evidence to demonstrate this effect.

Qualitative Assessment

I found the paper interesting to read and the theoretical analysis illuminating. It contributes to the understanding of dropout and provides a rather simple improvement of this very important technique with moderate gains. The main weakness of the paper are the empirical evaluation which lacks some rigor, and the presentation thereof: - First off: The plots are terrible. They are too small, the colors are hard to distinguish (e.g. pink vs red), the axis are poorly labeled (what "error"?), and the labels are visually too similar (s-dropout(tr) vs e-dropout(tr)). These plots are the main presentation of the experimental results and should be much clearer. This is also the reason I rated the clarity as "sub-standard". - The results comparing standard- vs. evolutional dropout on shallow models should be presented as a mean over many runs (at least 10), ideally with error-bars. The plotted curves are obviously from single runs, and might be subject to significant fluctuations. Also the models are small, so there really is no excuse for not providing statistics. - I'd like to know the final used learning rates for the deep models (particularly CIFAR-10 and CIFAR-100). Because the authors only searched 4 different learning rates, and if the optimal learning rate for the baseline was outside the tested interval that could spoil the results. Another remark: - In my opinion the claim about evolutional dropout addresses the internal covariate shift is very limited: it can only increase the variance of some low-variance units. Batch Normalization on the other hand standardizes the variance and centers the activation. These limitations should be discussed explicitly. Minor: *

Confidence in this Review

2-Confident (read it all; understood it all reasonably well)